# X-Intersected Silicon Modulator of Well-Rounded Performance

**DOI:** 10.3390/mi14071435

**Published:** 2023-07-17

**Authors:** Zijian Zhu, Yingxuan Zhao, Zhen Sheng, Fuwan Gan

**Affiliations:** 1National Key Laboratory of Materials for Integrated Circuits, Shanghai Institute of Microsystem and Information Technology, Chinese Academy of Sciences, 865 Changning Road, Shanghai 200050, China; zhuzj@mail.sim.ac.cn (Z.Z.); zsheng@mail.sim.ac.cn (Z.S.); 2University of Chinese Academy of Sciences, Beijing 100049, China

**Keywords:** silicon modulator, 3D doping design, modulation efficiency, carrier-induced loss, electro-optic bandwidth

## Abstract

In silicon modulator design, implantation is always a key factor, significantly influencing the doping profile and carrier distribution. As waveguide doping is limited by the compact footprint of the modulator rib, three-dimensional complex optimization is a viable option to improve performance. This work proposes an X-intersected modulator based on two inversely slanted junctions using the effective 3D Monte Carlo method for junction generation. The optimized results show that the modulation efficiency of the design is 1.09 V·cm, while the loss is 18 dB/cm, and the 3 dB bandwidth reaches over 35 GHz owing to the decreased resistance and capacitance of the 3D junction. This work demonstrates the benefits of 3D doping design in silicon modulators, contributing to higher efficiency and avoiding additional PN overlap to introduce lower capacitance. The design of 3D doping profiles well balances the DC and AC performance, and provides novel modulator solutions for high-speed datacom.

## 1. Introduction

Demand for data traffic has reached unprecedented levels nowadays. From long-haul telecommunications to on-chip interconnects, the requirement for high-speed communications is paramount. In this scenario, optical links emerge as the ideal solution. Silicon photonics (SiP), because of its compatibility with a complementary metal-oxide-semiconductor (CMOS) and its great potential for large-scale integration, has been widely applied in optoelectronic communications [1]. Due to their compact size, mature manufacturing, and low cost, silicon modulators have been a key enabler of current and future high-speed data transmission [2,3,4]. For silicon modulators, due to the centrosymmetric structure of silicon, the mechanism of modulation is based on the plasma dispersion effect, which modulates the effective refractive index of silicon by controlling the density of free carriers under an external electric field [5]. Among different types of silicon modulators, the carrier-depletion type is the preferred choice in terms of balanced performance for high-speed applications [6]. However, carrier-depletion modulators suffer from low modulation efficiency, resulting in the requirement of high half-wave voltage ($V_{\pi }$) to drive the entire device. This limitation results in high power consumption and large device footprints, which are incompatible with the requirements of chip-scale photonics. As a result, modulator improvements that offer higher modulation efficiency have been under continuous investigation.

Since the inherent principle of modulation in silicon photonics is strongly related to carrier distribution, doping profiles have a major impact on the efficiency, insertion loss, and bandwidth of the modulator. Much work has been conducted to increase the variability of the effective index under bias changes, without significantly increasing the capacity [7]. The main focus of previous research has been the modification of the doping profile in the ridge waveguide cross-section. However, the compact size of the ridge waveguide limits the further optimization other than the lateral junction [8], the vertical junction [9], the L-shaped junction [10], the U-shaped junction [11], and the wrapped junction [12]. Focusing on a new dimension of optimization, the creation of junctions in the direction of light propagation, has proven to be very positive in improving modulation efficiency and reducing propagation losses [13,14,15,16,17]. In modulators based on interleaved PN junctions, the overlap between the P-type and N-type regions at the interface is remarkably strong. This strong overlap results in a large extension of the depletion region, and consequently, the overlap between the optical mode and the depletion region is significantly enlarged, which leads to efficient modulation. Moreover, lower doping levels are required in interleaved modulators to achieve high modulation efficiency, so the carrier-induced loss is also reduced. Nevertheless, the interleaved combinations of pure p-type and pure n-type implantation have been trapped in high capacitance due to the large area of the PN junction, while the PN overlap in the corners of the cross-section contributes little to improving the modulation efficiency [18,19,20]. Therefore, the improvement of the frequency response is the limitation of the interleaved modulator and its use in high rate applications. The dilemma of cross-section and interleaved optimization inspires that the design of complex doping profiles in three dimensions has enormous potential for comprehensive high-performance modulation.

Exploiting the 3D design idea, this paper proposes a potent silicon Mach–Zehnder modulator (MZM) based on two inversely slanted junctions, which is named the X-intersected modulator. Based on the effective 3D Monte-Carlo method, the modulator offers the modulation efficiency of 1.09 V·cm and the insertion loss of 18 dB/cm. The RC parameters of the modulator are also at a low level, boosting the electro-optic (EO) bandwidth to over 35 GHz, which permits high modulation speed. The benefits of the 3D doping profile for silicon modulators are revealed, making the 3D-junction modulator a clear contender for high-baud optical links at present and in the future.

## 2. Methods

To obtain a well-performing modulator, both the optical and electrical structures must be carefully considered. The structure of the silicon MZM with the transmission line is shown in Figure 1a, where one arm is doped for light modulation. Figure 1b shows the TE mode profile of the light propagating into the modulator waveguide as the footprint of the waveguide cross-section satisfies the single mode condition. The detailed waveguide cross-section design in Figure 1c specifies the rib height $H_{r}$ of 220 nm, the rib width $W_{r}$ of 450 nm, and the slab height $H_{s}$ of 60 nm. Based on the idea of leveraging complex 3D doping profiles, the 3D junctions should be formed as many as possible. Therefore, we have the periodic doping structure in Figure 1a. The top view of a single period is depicted in Figure 1d. Two different P-type and N-type implantation groups form the 3D doping profile at the interface of two groups. The junction of the 3D profile leads to the expansion of the depletion region both in the cross-section and in the propagation direction under modulation voltages, which is called the omni-junction and is advantageous for strong modulation. The two types of implantation groups occupy $L_{p1}$ and $L_{p2}$, respectively, in the Z direction. In this case, we have $L_{p1}=L_{p2}=600$ nm to make good use of space and take full advantage of the 3D junction. The entire length $L_{p}=L_{p1}+L_{p2}$ of the unit is adjustable to suit different requirements. For less simulation time, the simulation length of the unit period employs $L_{p}/2=L_{p1}/2$+$L_{p2}/2$, owing to the symmetry in the repetitive structure and ignorable carrier diffusion at simulation boundaries. The implantation steps follow the ordinary two-step fabrication process.

As implantation varies in 3D space, modeling complex 3D doping profiles is essential. The Monte-Carlo method is well known for accurately simulating ion implantation [21]. However, this method is usually applied under 2D conditions, as adding another dimension consumes massive computational time and resources. Therefore, in our previous work [22], we proposed the effective 3D Monte-Carlo method to generate 3D doping profiles while ensuring accuracy. The effective 3D Monte-Carlo method is enabled by projecting the 2D Monte-Carlo doping profiles in three dimensions with respect to areas inside or outside the implantation boundaries. The effectiveness of the model is verified by comparing the modeling results with experimental results of structures in [13,15]. With this modeling tool, any 3D doping profile is possible to analyze.

Since each implantation is influenced by the doses, energies, and locations, for two groups of implantation in the two-step fabrication process, 20 doping parameters need to be adjusted to seek the profile designs of great performance. The large quantity of optimization parameters presents challenges to optimization complexity. Thus, an efficient optimization method is required to lead doping parameters to converge to optimal combinations and obtain ideal results fast. In this work, the particle swarm optimization (PSO) algorithm is adopted to find suitable doping parameters because PSO is favorable for fast convergence and insensitivity to scaling of design variables [23,24]. By setting an appropriate target, the figure of merit (FOM), PSO algorithm will work accurately and efficiently. The FOM needs to consider all aspects of vital indices of a silicon modulator and thus is defined as:(1)$$FOM=\theta \cdot V_{\pi }L+\frac{V_{\pi }L\cdot Loss}{R\cdot C^{2}}$$
where $V_{\pi }$ is the half-wave voltage at the bias of −2 V. *L* is the length of the active region of the modulator arm. Loss is the insertion loss at the bias of 0 V, where the carrier-induced loss is mainly considered propagation loss and is comparatively much smaller in the silicon waveguide. *R* and *C* are the resistance and capacitance of the PN junction at the bias of −2 V, reflecting the variation of EO bandwidth. θ is the weight of $V_{\pi }L$ to emphasize the modulation efficiency of the design since the latter part of the FOM is affected by the bandwidth to a large extent.

The block diagram of the process in each iteration of the PSO algorithm is shown in Figure 2. In each iteration, different groups of particles have different implantation parameters. For each particle group, the doping profile is generated by the 3D Monte-Carlo method based on the group parameters. Once the doping profile has been established, DC simulation is performed by TCAD tools to obtain the change of the electron and hole distribution under bias, and the AC simulation is carried out to obtain the CG parameters (capacitance and conductance) of the entire PN junction and thus solve the RC parameters. With the carrier distribution variation, the change in the effective index is modeled according to the plasma dispersion effect and thus the $V_{\pi }L$ and loss can be solved from the real part and the imaginary part of index changes. Using the RC parameters of the PN junction and the segmented transmission line that is generally preset, the EO bandwidth is estimated in accordance with the transmission line theory. Notably, the final segmented transmission line for the obtained optimum doping profile has optimized electrode spacing and distances for improved microwave performance such as reduced microwave attenuation, better impedance matching, and better microwave index matching. The detailed calculations of $V_{\pi }L$, loss, and bandwidth are included in Section 3. Then, the FOM of each particle group is acquired to symbolize the overall performance of the modulator with certain implantation parameters. By comparing these FOMs, the optimization directions of the particle groups are guided in the next iteration. The whole optimization is iterated in this process and after the last iteration, we analyze the output results and focus on optimal and practical solutions.

## 3. Result and Discussion

Via the PSO algorithm with particle groups of 20 [25], iteration steps of 30 and θ=100 in FOM to balance weights of performance, the algorithm fitness versus iterations is plotted in Figure 3. The iteration of 30 satisfies the convergence of optimization. The fluctuations in the curve are caused by the fact that small changes in implantation parameters that possibly affect the FOM of the modulator to an enormous degree. Therefore, an appropriate FOM is extremely important for optimization guidance, and the FOM in this work is examined to be effective. With 30 times iterations and 20 particle groups, the entire optimization process, including 3D Monte-Carlo generation of the doping profile, calculation of DC performance ($V_{\pi }L$, Loss), and estimation of the AC response (RC parameters, EO bandwidth), takes between 15 min and 1 h to simulate each particle group, so the total optimization time is approximately two weeks. The process time is the result of a trade-off between simulation accuracy, output diversity, and convergence speed. After analyzing the optimized results, the optimal and practical doping profile is denoted by Figure 4. Figure 4a visualizes the schematic doping profile of a single doping period composed of two types of junctions, which are respectively depicted in Figure 4b,c. The optimized implantation parameters of two junctions are shown in Table 1 and Table 2. The implantation is workable as the parameters are within the scope of the fab process and the fabrication steps in the tables allow experimental achievement. The implantation is injected vertically and the tilt angle is 0°. The rapid thermal annealing for the model is 1030 °C within 10 s, which is estimated from the commercial fab process. The estimation is based on the fitting to the sheet resistance of the P/N doping provided by the fab handbook, which has multiple solutions and the annealing temperature and time adopt the values in the range of 850∼1200 °C and 0∼100 s [26]. While for other known and accessible annealing conditions, the simulation method is still adaptable and the 3D doping profile in this work is feasible by adjusting the implantation conditions utilizing the PSO algorithm. Doping levels of P-type and N-type implantation are both around $1\times 10^{18}$/cm${}^{3}$ to balance the variation of the effective index and the carrier-induced loss. The two junctions are slanted in opposite directions resembling ’X-intersected’.

With the finite difference eigenmode (FDE) solver in the Lumerical software 2020 R2.4, the modulation efficiency and carrier-induced loss are plotted in Figure 5 and Figure 6. The solid lines indicate the variation of the values along the propagation direction. The deducing details are shown below. For the point *z* in the propagation direction, the phase difference dϕ(z) under the bias $V_{b}$ is:(2)$$d\phi \left(z\right)=\frac{2\pi \Delta n(z)}{\lambda }dz$$
where Δn(z) is the change of the real part of the effective index, λ is the optical wavelength. The half-wave voltage-length product $V_{\pi }L_{z}$ at *z* is calculated as:(3)$$V_{\pi }L_{z}=V_{pp}L_{\pi }=\frac{V_{pp}\lambda }{2\Delta n(z)}$$

The carrier-induced loss $Loss_{z}$ is obtained from the imaginary part of the effective index k(z), as:(4)$$Loss_{z}=10lg\left(e\right)\frac{4\pi k(z)}{\lambda }$$

The dashed lines are the overall performance of the period. The overall phase difference $\Delta \phi_{p}$ of one period is the integration of dϕ(z):(5)$$\Delta \phi_{p}=\int d\phi \left(z\right)$$

The overall $V_{\pi }L$ of the period with the length *L* becomes: (6)$$V_{\pi }L=\frac{V_{pp}\pi L}{\Delta \phi_{p}}$$

The overall loss is the average of $Loss_{z}$ over the period.

At the bias from −1 V to −3 V and peak-to-peak voltage ($V_{pp}$) of 2 V, $V_{\pi }L$ ranged from 0.81 V·cm to 1.33 V·cm, specifically being 1.09 V·cm at −2 V. The insertion loss of the period was estimated to be 18 dB/cm, while the loss was reduced to 14.3 dB/cm at −2 V. Low $V_{\pi }L$ was attributed to both the 3D junction and the well-designed former junction. The former junction provides much lower $V_{\pi }L$, as illustrated by Figure 5 in the range of 0∼0.3 μm. Apart from the contribution from the former junction, the valleys on the solid lines also help decrease $V_{\pi }L$, which is the effect of the 3D junction. When biased at −1 V, the valleys locate 0.02∼0.08 μm away from the interface at the side of the former doping and 0.03∼0.15 μm away from the interface at the side of the latter doping. The peak on the side of the latter half near the interface is due to the fact that the depletion region generated at the interface when no bias was applied is mainly extended to the latter half. With higher bias, the valleys become shallower and the peak becomes more towering, which is consistent with the actual situation as the depletion region keeps extending, resulting in a wider junction with fewer carriers. Therefore, this type of modulator with the 3D junction is more suited to low-voltage applications. This specificity is also in line with the development trend. A noticeable phenomenon is that the former junction increases more in $V_{\pi }L$ and decreases more in loss under bias compared to the latter junction. The reason is that the former part takes the form of an L-shaped junction, which allows the depletion region to expand faster and larger across the mode center under bias. Therefore, as the bias increases, the former junction will have a higher downward gradient of carrier distribution change and, therefore, a higher sensitivity of the bias response.

The loss in Figure 6 displays opposite locations of valleys and peaks compared to the $V_{\pi }L$. Since the depletion region has fewer carriers, in terms of $\Delta \alpha =8.5\times 10^{-18}\times \Delta n_{e}+6.0\times 10^{-18}\Delta n_{h}$, the loss is declined. From $V_{\pi }L_{z}$ and $Loss_{z}$, two junctions complement each other in modulation efficiency and loss. Therefore, composing a 3D doping profile, not only provides the 3D junction for higher modulation efficiency, but also combines different advantages of the two PN junctions, thus delivering high-level DC performance.

The AC performance of the modulator depends on the resistance and capacitance of the PN junction. The circuit models of a 2D PN junction and a 3D PN junction by the transmission line are shown in Figure 7. $R_{e},L_{e},C_{e},G_{e}$ are the parameters of the transmission line. $C_{pn}$ is the capacitance of the junction. In the 2D PN junction, the current has only one path choice from the N-type doped side to the P-type doped side, hence the resistance is $R_{c}$ in Figure 7a. While in the 3D PN junction, the carriers are driven both in the cross-section and in the propagation direction; therefore, the resistance in Figure 7b has two parallel parts, $R_{c}$ for the transverse resistance in the cross-section and $R_{p}$ for the resistance in the propagation direction. For this design, the calculated $R_{c}$ at 0 V reaches over 200 kohm·μm, which is adverse to high-speed applications, while $R_{p}$ provides an alternative path for the current, leading to low resistance, as denoted by Figure 8a. The capacitance of the period per unit length is shown in Figure 8b. The capacitance of the X-intersected junction compared with other reported interleaved PN junctions is much lower for the appropriate overlap of two doping profiles at the interface in this design. The area of the PN overlap in interleaved PN junctions takes up the whole rib cross-section, thus it has extremely high capacitance. On the contrary, this 3D doping profile has PN overlap surrounding the optical mode center to avoid introducing excess capacitance, while maintaining effective modulation.

The design of the electrode is based on the segmented transmission line to reduce the microwave velocity and the impedance for higher bandwidth. Figure 9 shows the structure of the segmented transmission line, which is accessible in the CMOS foundry. $L_{seg}$ is the segment length of 8.4 μm. $L_{int}$ is the interval between adjacent segment electrodes, 1.6 μm. $G_{m}$ is the gap between metal lines, 8 μm.

Given the resistance and capacitance of the period and the segmented transmission line structure, the bandwidth of the designed modulator was estimated using microwave transmission (ABCD) matrix theory [27].
(7)$$S_{21}=\left|\frac{\Sigma V_{j}}{V_{pp}/2\times \Sigma L_{seg}\times \left(1+j\omega R_{pn}C_{pn}\right)}\right|$$
where $V_{j}$ is the voltage distributed onto the 3D PN junction [28], which has impedance mismatch, microwave loss, microwave dispersion effects, and velocity mismatch taken into consideration. $R_{pn}$ and $C_{pn}$ are the resistance and capacitance of the modulator period. Based on the ABCD matrix, the EO frequency response of the modulator with 1 mm length is calculated in Figure 10. At the bias of −2 V, the 3 dB EO bandwidth reaches over 28 GHz and is over 35 GHz at −3 V. The high bandwidth verifies that this 3D-junction modulator offers better serviceability for high-speed requirements, compared with other reported interleaved modulators.

The fabrication tolerance of the X-intersected modulator is analyzed in Figure 11, taking the drift errors of former and latter junctions into account. In the analysis, the implantation location error of the slanted junction rightwards is labeled as positive and the location error leftwards is labeled as negative. The error is within the range of 50 nm. From Figure 11a,d, the $V_{\pi }L$ variation is below 5%. The loss variation is below 2%. Therefore, the modulator exhibits robust DC performance. The bandwidth fluctuates slightly with the drift error of the latter implantation. However, the former implantation location has a strong effect on bandwidth, which is avoidable by designing a leftward implantation margin. In this case, the fabrication tolerance of the X-intersected modulator is adequate for manufacturing.

The comparison between the designed X-intersected modulator with other reported interleaved PN modulators is demonstrated in Table 3. $V_{\pi }L\times$ Loss is applied to compare the DC performance. The design in this work provides low $V_{\pi }L$ with a doping level of $1\times 10^{18}$/cm${}^{3}$, which can be increased to $5\times 10^{18}$/cm${}^{3}$ in [20] to obtain much lower $V_{\pi }L$, yet sacrificing the carrier-induced loss. The great improvement of our work is the bandwidth broadening, which is raised up to 35 GHz, addressing the problem of high capacitance in the interleaved PN junctions. The wider EO bandwidth is due to the appropriate PN overlap concentrated around the center of the optical mode, enabled by this X-intersected doping design. In addition, this 3D junction allows for more conductive paths, which reduces the junction resistance and is beneficial for achieving higher baud rates. The comparison elaborates on the superiority of the complex 3D doping profiles for silicon modulators.

## 4. Conclusions

In conclusion, this paper proposes an X-intersected silicon modulator of high performance. By composing an effective 3D profile, the modulator is enabled with high EO bandwidth of 35 GHz with the $V_{\pi }L$ of 1.09 V·cm and loss of 18 dB/cm. The modulator also shows robustness both for DC and AC performance. This work reveals that the 3D doping profile not only provides the 3D junction for high modulation efficiency and low loss, but also declines the resistance and capacitance of the modulator. The 3D doping design is beneficial for silicon modulators in all aspects, and thus leverages modulator performance to the next level. The 3D idea is also applicable to microring modulators, which remains as our future work, together with more efficient profile designs.

## Figures and Tables

**Figure 1 micromachines-14-01435-f001:**
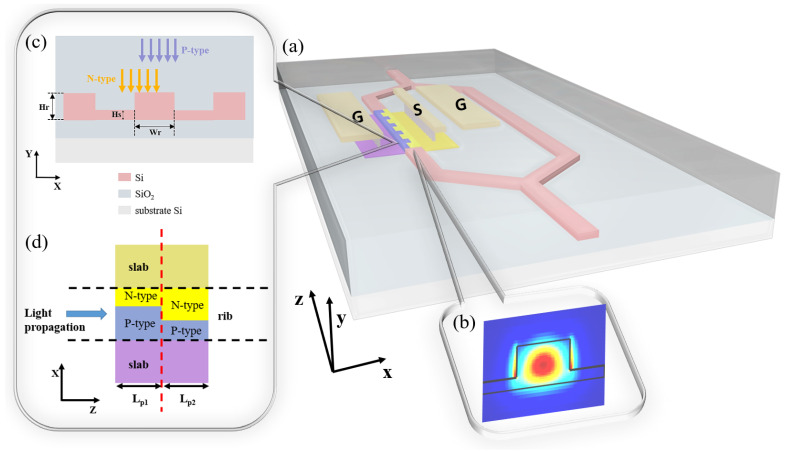
(**a**) The top view of the 3D MZM consisting of periods whose lengths are all $L_{p}$. (**b**) The optical mode profile of the rib modulator. (**c**) Cross-section of the modulator with two-step doping process. (**d**) The top view of the modulator period consisting of two doping parts.

**Figure 2 micromachines-14-01435-f002:**
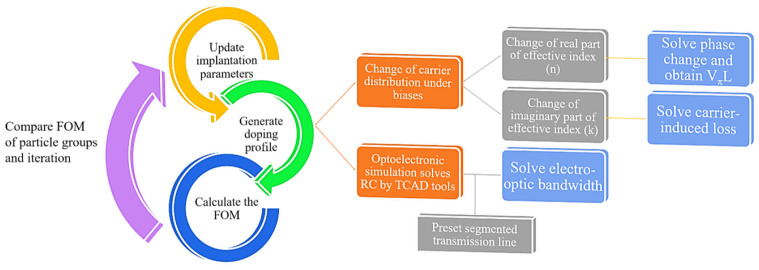
The block diagram of the modulator optimization pipeline.

**Figure 3 micromachines-14-01435-f003:**
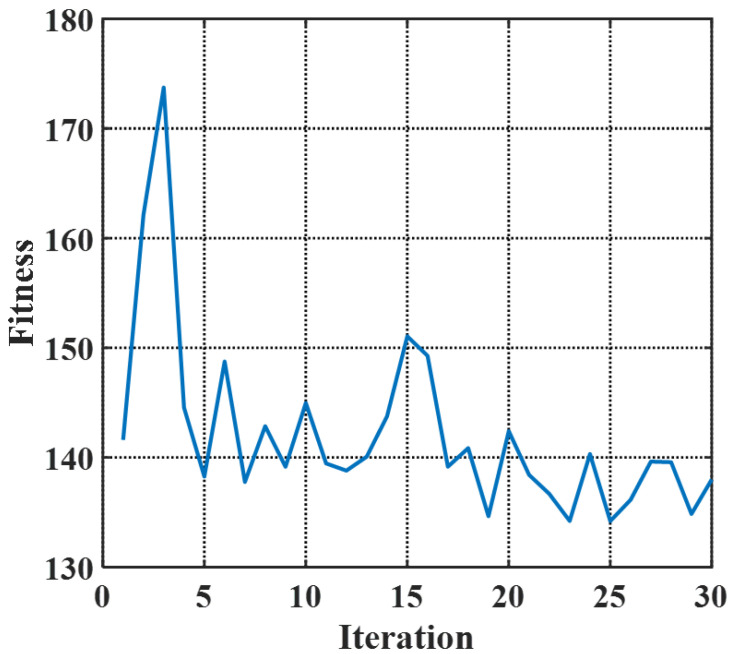
The plot of the algorithm fitness versus iterations.

**Figure 4 micromachines-14-01435-f004:**
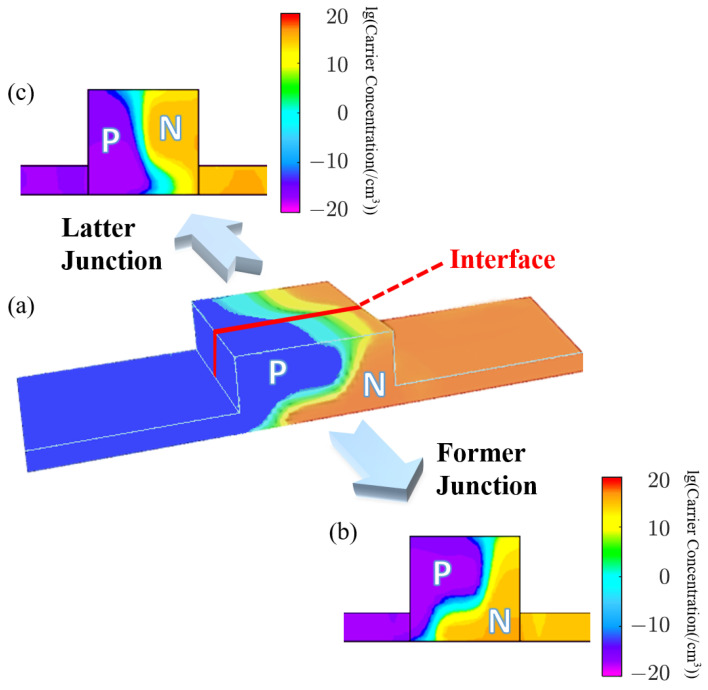
(**a**) Schematic diagram of the omni-junction modulator based on vertical and lateral junctions; (**b**) lateral and (**c**) vertical junctions of a single period.

**Figure 5 micromachines-14-01435-f005:**
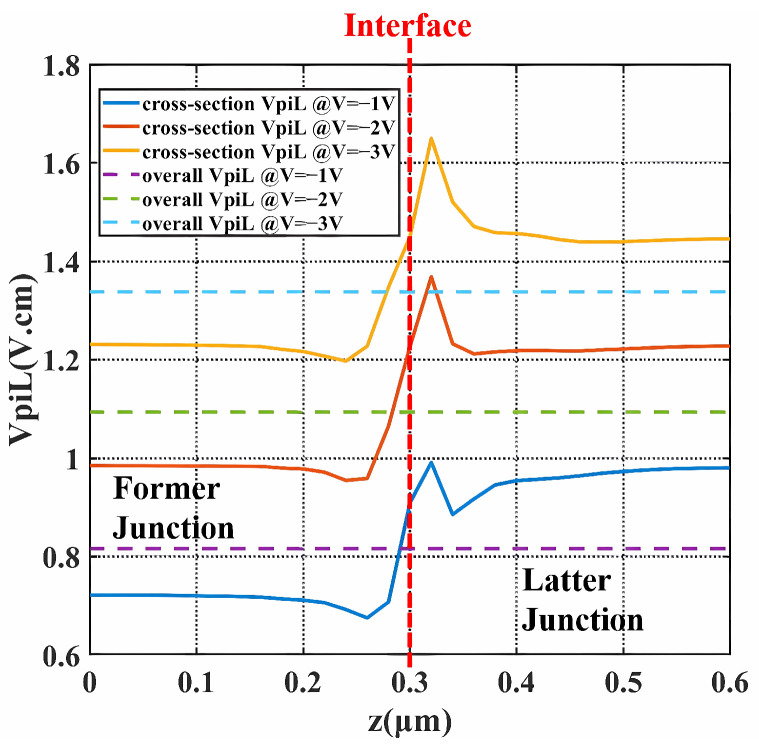
$V_{\pi }L$ of the X-intersected modulator over the propagation direction under bias. (solid lines are the $V_{\pi }L_{z}$ of each cross-section along the Z direction and dashed lines are the overall $V_{\pi }L$ of the whole period).

**Figure 6 micromachines-14-01435-f006:**
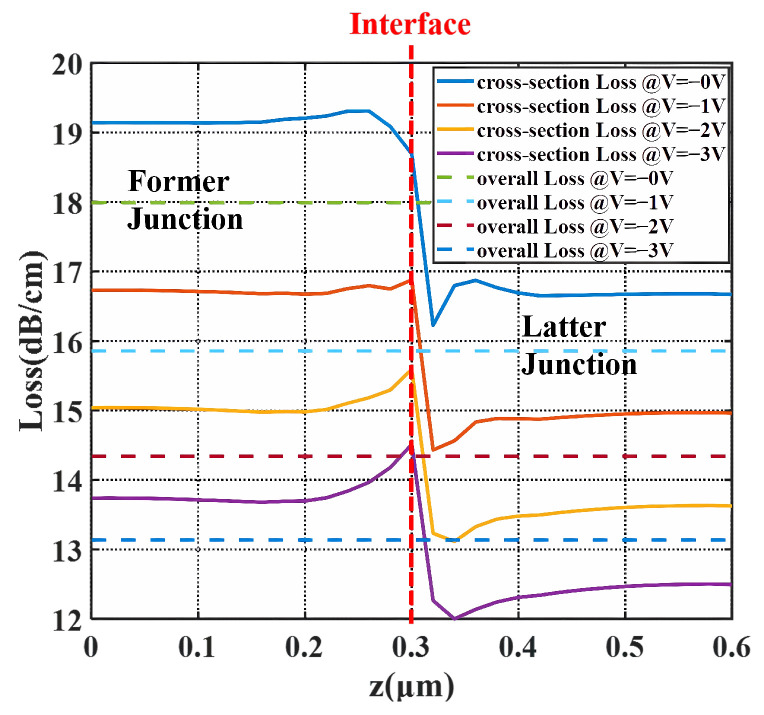
Carrier-induced loss of the X-intersected modulator over the propagation direction under bias. (Solid lines are the $Loss_{z}$ of each cross-section along the Z direction and dashed lines are the overall loss of the whole period.)

**Figure 7 micromachines-14-01435-f007:**
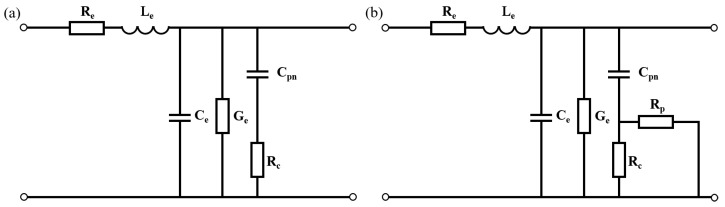
Circuit models of (**a**) the 2D PN junction modulator and (**b**) the 3D PN junction modulator.

**Figure 8 micromachines-14-01435-f008:**
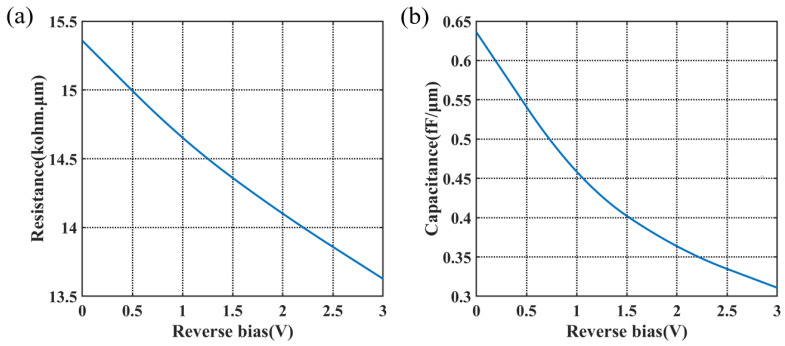
(**a**) Resistance and (**b**) capacitance per unit length of the modulator period.

**Figure 9 micromachines-14-01435-f009:**
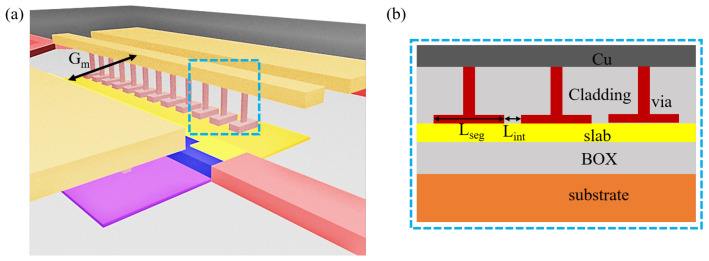
(**a**) The schematic diagram and (**b**) the detailed side view of the segmented transmission line.

**Figure 10 micromachines-14-01435-f010:**
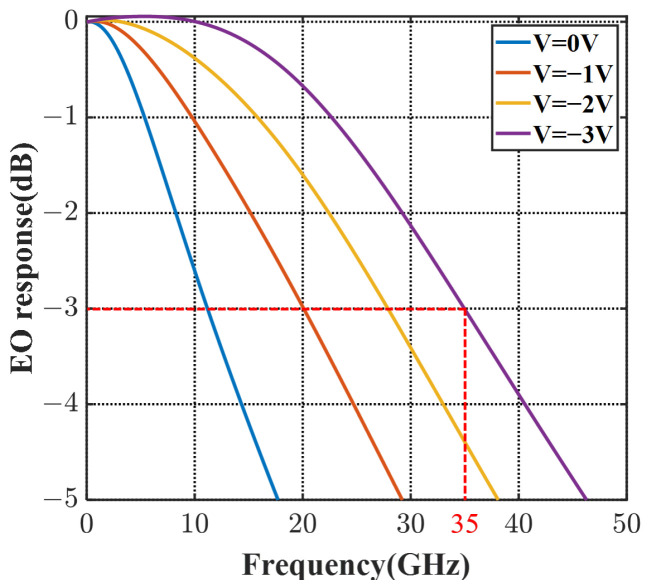
The frequency response of the X-intersected modulator at 0∼−3 V bias.

**Figure 11 micromachines-14-01435-f011:**
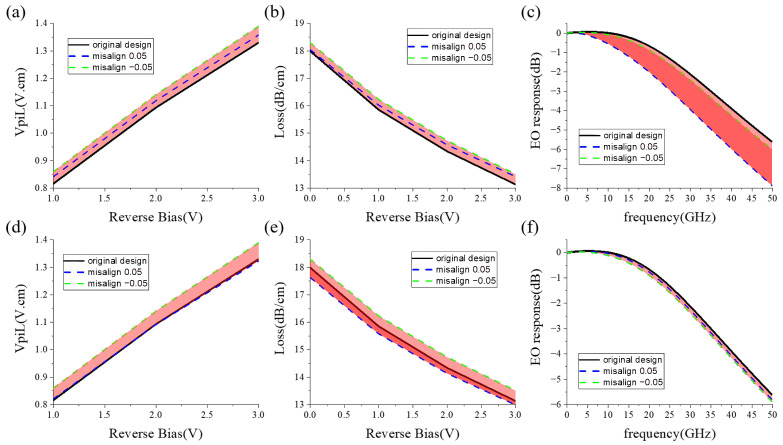
The fab tolerance of the X-intersected modulator: (**a**) $V_{\pi }L$, (**b**) carrier-induced loss, (**c**) bandwidth influenced by the location error of the former implantation, (**d**) $V_{\pi }L$, (**e**) carrier-induced loss, (**f**) bandwidth influenced by the location error of the latter implantation. (Black lines depict the performance of the original X-intersected modulator. Blue and green dashed lines are the performance of the modulators with positive and negative misalignment, respectively.)

**Table 1 micromachines-14-01435-t001:** Implantation parameters of the junction in Figure 4b.

Steps	Species	Dose (cm${}^{-2}$)	Energy (keV)	Window ${}^{a}$ (μm)
1	Phosphorus	$2\times 10^{13}$	120	(−0.126, 3)
2	Phosphorus	$2\times 10^{13}$	130	(−0.126, 3)
3	Boron	$1\times 10^{13}$	50	(−3, 0.074)
4	Boron	$2\times 10^{13}$	20	(−3, 0.074)

${}^{a}$ The window is based on the coordinate system with the origin at the rib center.

**Table 2 micromachines-14-01435-t002:** Implantation parameters of the junction in Figure 4c.

Steps	Species	Dose (cm${}^{-2}$)	Energy (keV)	Window ${}^{a}$ (μm)
1	Phosphorus	$2\times 10^{13}$	140	(0.011, 3)
2	Phosphorus	$2\times 10^{13}$	60	(0.011, 3)
3	Boron	$2\times 10^{13}$	40	(−3, 0.205)
4	Boron	$1\times 10^{13}$	45	(−3, 0.205)

${}^{a}$ The window is based on the coordinate system with the origin at the rib center.

**Table 3 micromachines-14-01435-t003:** Comparison with reported C-band high-speed modulators optimized in the propagation direction.

Property	Year	$V_{\pi }L$ (V·cm)	Loss (dB/cm)	$V_{\pi }L\;\times \;$ Loss (V·dB)	Bandwidth (GHz)
[14]	2011	2.5@−2 V	34	85	N.A. (10 Gbps)
[13]	2012	1.7@−2 V	10	17	20@−3 V
[15]	2013	2.4	21	50.4	20@−3 V
[17]	2018	2.4	N.A.	N.A.	34@−3 V
[20]	2022	0.23@−2 V	63	14.49	6.8@−1 V
This work	2023	1.09@−2 V	18	19.62	35@−3 V

## Data Availability

The data presented in this study may be available from the corresponding author upon reasonable request.

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
