# Peer review of "X-Intersected Silicon Modulator of Well-Rounded Performance"

_micromachines, 2023, doi:10.3390/mi14071435_

Round 1
Reviewer 1 Report
The present manuscript entitled’’X-intersected Silicon Modulator of Well-rounded Performance’’ by Zhu et al. has designed a silicon modulator, using the 3D design idea. In this modulator, two inversely slanted junction design is shown to high modulation efficiency with insertion loss of 18 dB/cm. This model shows a high modulation speed. 2D Monte Carlo method is modified to a 3D Monte-Carlo method as a modeling tool. Furthermore, implantation experimental conditions are investigated. These findings of this research work open an avenue of using a 2D model as effective 3D model (already published by same authors). The authors report interesting work. The objective and justification of the work reported is clear. However, some issues are detailed below which need to be addressed before its final acceptance in Micromachines.
I advise the authors to take the following points into account while revising their manuscript.
1. There are some typographical and grammatical errors in the manuscript text, so the authors need to correct them in the revised manuscript and correct the subscript errors. E.g., Line 43, reference.
2. Authors use a 3D effective model; what accuracy can be quantified here?
3. As 3D doping designs are beneficial for silicon modulators suggested by authors, I wonder if this is complex to achieve experimentally.
4. In comparison to Table 3, one reference is symbolically ‘?’. please correct.
5. Authors suggest an implantation condition of the rapid thermal annealing for the model is at 1030°C within 10 s. This cannot be supported practically. What is the author's opinion on this?
Moderate editing is required. Scientific soundness is clear.
Author Response
Thank you for your kind comments. We have responded to the comments and modified our paper for a better presentation. Please see the attachment.

Reviewer 2 Report
In this manuscript, the authors present the design of a silicon electro-optic modulator with a 3D doping profile. The dopant implantation process is simulated using the 3D Monte Carlo method, and the doping profile is meticulously optimized to enhance the figure-of-merit. The modulator's performance is evaluated based on modulation efficiency, insertion loss, and bandwidth. By comparing the results with interleaved pn modulators, the authors highlight numerous advantages, particularly in terms of bandwidth. This work is intriguing and demonstrates a promising strategy for augmenting the performance of on-chip electro-optic modulators. I recommend its publication, but there are a few concerns that should be addressed beforehand.
1) The pipeline of the simulation should be presented explicitly. How are the parameters (V_pi, R, C, loss, etc..) calculated in each iteration? The authors should consider presenting a block diagram illustrating the optimization process.
2) The manuscript mentions that the optimization is fixed with 30 iterations. It would be valuable to know how quickly the optimization converges within these iterations. The authors can provide a plot of Figure-of-Merit versus iteration to demonstrate the convergence. Additionally, it would be beneficial to mention the time taken for the entire optimization process.
3) Could the modulator’s performance be further enhanced by using a large unit cell (larger Lp) in the optimization? Please comment on the simulation constraint on Lp.
4) What is the interpretation for the observed higher sensitivity of the response to the former junction compared to the latter junction?
5) Several citation numbers are missing. (line 43, 187, Table 3)
Author Response

(The authors gave the same response as above.)
